# Analysis of Teachers’ Visual Behaviour in Classes: A Systematic Review

**DOI:** 10.3390/ejihpe15040054

**Published:** 2025-04-05

**Authors:** Rodrigo Mendes, Mário Pereira, Paulo Nobre, Gonçalo Dias

**Affiliations:** 1Faculty of Sports Sciences and Physical Education, University of Coimbra, 3040-248 Coimbra, Portugal; uc48493@uc.pt (M.P.); prnobre@fcdef.uc.pt (P.N.); 2Coimbra Education School, Polytechnic Institute of Coimbra, Rua Dom João III-Solum, 3030-329 Coimbra, Portugal; goncalodias@esec.pt; 3Laboratório RoboCorp, IIA, Instituto Politécnico de Coimbra, 3040-256 Coimbra, Portugal; 4Applied Sport Sciences Research Unit (UNICID-ASSERT), Coimbra Education School, Instituto Politécnico de Coimbra, 3040-256 Coimbra, Portugal; 5Interdisciplinary Centre for the Study of Human Performance (CIPER), Faculty of Human Kinetics, University of Lisbon, 1495-751 Cruz Quebrada-Dafundo, Portugal; 6Center for Interdisciplinary Studies (CEIS20), University of Coimbra, 3030-789 Coimbra, Portugal; 7Sport Physical Activity and Health Research & Innovation Center (SPRINT), 3030-329 Coimbra, Portugal

**Keywords:** education, eye tracking, learning, student performance, teaching, visual behaviour

## Abstract

(1) Background: Teachers’ visual behaviour in classes has an important role in learning and instruction. Hence, understanding the dynamics of classroom interactions is fundamental in educational research. As mapping evidence on this topic would highlight concepts and knowledge gaps in this area, this systematic review aimed to collect and systematise the analysis of teachers’ visual behaviour in classroom settings through the use of eye-tracking apparatus; (2) Methods: The methodological procedures were registered in the INPLASY database and this systematic review used the PRISMA criteria for the selection and analysis of studies in this area. We searched on six literature databases (B-on, ERIC, ScienceDirect, Scopus, TRC and WoS) between 1 January 2015 and 31 December 2024. Eligible articles used eye tracking apparatus and analysed teachers’ visual behaviour as a dependent variable in the experiment; (3) Results: The main results of the articles selected (*n* = 41) points to the differences in teachers’ visual behaviour in terms of professional experience and the relationship between gaze patterns and several classroom variables; (4) Conclusions: A deeper understanding of teachers’ visual behaviour can lead to more effective teacher training and better classroom environments. The scientific research in this area would benefit from more standardized and robust methodologies that allow more reliable analyses of the added value of eye tracking technology.

## 1. Introduction

Understanding the dynamics of classroom interactions is fundamental in educational research, particularly in the teaching effectiveness’ and student engagement’s context. One critical dimension of this dynamic is the teachers’ visual behaviour during educational activities ([3]).

Teachers’ gaze patterns offer valuable insights into their instructional focus and responsiveness to students’ needs, as well as classroom management strategies. The emergence of eye-tracking technology has provided an innovative means to objectively measure and analyse these visual behaviours, offering unprecedented depth and precision to understand how teachers visually interact with the classroom environments ([25]).

Eye-tracking technology has acquired considerable relevance in educational settings. By capturing and quantifying eye movements, saccades and fixations, eye-tracking apparatus enables scientific researchers to decode visual attention and cognitive processes ([1]).

This methodological advancement has considerable implications for educational investigation, particularly in examining how teachers allocate their visual attention across the various classroom stimuli, such as students, instructional materials and classroom management tasks, making eye tracking a powerful tool for studying teachers’ decision-making processes and how they process classroom behaviour ([2]). These analyses can illuminate the cognitive underpinnings of teaching practices and potentially inform strategies for teacher training and professional development ([20]).

Teachers’ visual behaviour has an important role in learning and instruction ([6]). Visual attention is integral to how teachers navigate complex classroom environments. Additionally, gaze has a communicative and social function in interactions, suggesting that how and where teachers look can influence classroom dynamics and student engagement. By integrating these ideas, the analysis of teachers’ gaze patterns through eye-tracking technology can offer comprehensive insights into teaching’s multifaceted nature ([43]).

The expanded relevance of eye-tracking research in pedagogy-oriented disciplines, where understanding visual attention can provide critical insights into teaching and learning processes, demonstrated how teachers’ and students’ gaze behaviour influences knowledge acquisition and classroom interactions. These investigations have revealed rich findings, such as how expert teachers distribute their attention differently than novices or how specific visual cues can enhance student engagement and comprehension ([15]; [35]).

Furthermore, eye-tracking studies in pedagogical contexts have explored the impact of teachers’ visual attention patterns in their students’ learning effectiveness and participation levels, as a stimulus and a key guide to the learners ([24]; [50]).

Despite the integration of eye-tracking technology in educational research offers a promising avenue for uncovering the complexities of teachers’ visual behaviour in classroom settings and the growing interest in this research domain, existing studies remain fragmented. Variations in research methodologies, analytical approaches or sample populations have resulted in a heterogeneous body of literature ([46]).

This variety makes it the development of a cohesive understanding of how teachers’ visual attention influences instructional effectiveness and students’ outcomes difficult. Moreover, discrepancies in experimental designs and the metrics analysed further complicate the synthesis of findings across studies. These methodological divergences underscore the need for a systematic review to aggregate the existing research ([1]).

A systematic review of teachers’ visual behaviour using eye-tracking apparatus is, therefore, both timely and necessary. Such a review of the emerging body of literature is essential to consolidate and provide a structured synthesis of existing knowledge, identifying and highlighting methodological trends and gaps on how eye-tracking technology has been applied to study teachers’ gaze patterns. It can also contribute to elucidate the methodological strengths and limitations in this field, guiding future research towards more robust and standardised scientific investigation approaches ([37]).

Additionally, by identifying consistent patterns in teachers’ visual behaviour, this review can provide evidence-based practices for teacher education and professional development, ultimately aiming to enhance, not only teaching effectiveness, but also student learning experiences ([30]).

By synthesising the main findings from the observational and experimental studies, this review seeks to examine how teachers distribute their visual attention, the factors influencing their gaze patterns and the implications of these behaviours for instructional practice, providing a comprehensive understanding of the current state of research in this field from the last ten years, offering insights that can bridge the gap between empirical evidence and practical application in educational contexts.

In view of the above, the aim of this systematic review is to collect and systematise the analysis of teachers’ visual behaviour in classroom settings through the use of eye-tracking technology apparatus.

## 2. Materials and Methods

### 2.1. Search Strategy

This systematic review used the “Preferred Reporting Items for Systematic Reviews and Meta-analysis” (PRISMA) to select and analyse the studies initially identified ([31]). The methodological procedures were registered in the “International Platform of Registered Systematic Review and Meta-analysis” database under the registration number INPLASY2024120086. As a result of the bibliographical research carried out and analysed, it was necessary to change the protocol once.

A search was conducted in six databases: Biblioteca do Conhecimento Online, Educational Resource Information Center, ScienceDirect, Scopus, Teacher Reference Center and Web of Science Core Collection. The terms (“eye tracking” OR “eye tracker”) AND (“teacher” OR “teaching”) AND (“visual behaviour” OR “visual focus” OR “visual attention” OR “eye gaze” OR “professional vision”) used were searched for in the titles, abstracts and keywords (Figure 1).

To broaden the spectrum of results ([1]; [46]), the time span of the research developed was between 1 January 2015 and 31 December 2024 (Figure 2). Afterwards, the main cross-references of the articles included in the systematic review were scanned.

### 2.2. Eligibility Criteria

The inclusion criteria in this review were the following: (i) published works between 1 January 2015 and 31 December 2024, (ii) works written in English, Portuguese, Spanish or French, (iii) articles published in peer-reviewed journals, (iv) articles in full-text, (v) research that used eye tracking apparatus and (vi) research where teachers’ visual behaviour was one of the dependent variables in the experiment.

These languages and the timespan of the research cover the articles that will be a reference in this area of knowledge and there is no reason to think that grey literature makes significant contributions to the review. Also, it was ensured that all included studies could be fully and critically analysed by the authors, minimising the risk of misinterpretation due to translation barriers.

The following criteria were used for exclusion: (i) published works outside the timespan selected, (ii) works written in languages other than those selected, (iii) academic theses, books, opinion articles, conference papers and non-scientific articles (iv) articles without full-text, (v) research that didn’t use eye tracking apparatus and (vi) research where eye tracking apparatus was used in the experiment for a purpose other than the intended (Appendix A).

The article selection process followed the following steps: (i) studies that used the descriptors in the aforementioned databases, (ii) exclusion of duplicate articles, (iii) reading of the titles, abstracts and keywords and (iv) critical reading and assessment of the articles.

The selection and extraction of data from the articles was carried out in three stages. First, two authors (R.M. and M.P.) independently selected eligible articles. Second, the same authors independently extracted the previously defined information. In both stages, a consensus of more than 90% was reached between these two authors, when comparing the analyses. If necessary, in case of disagreements, a third author (G.D.) was called to analyse and issue his final decision. Third, after achieving this homogeneity of criteria in more than 10% of the articles, one author (R.M.) completed the collection of the remaining eligible articles and information.

### 2.3. Quality Assessment

The “STrengthening the Reporting of OBservational studies in Epidemiology” (STROBE) tool was used by two authors (R.M. and M.P.) to independently assess the quality of the non-randomized studies included ([8]). Again, after achieving a homogeneity of criteria superior to 90% in more than 10% of eligible articles, one author (R.M.) completed the assessment.

This checklist aims to ensure a clear presentation of what was planned and conducted in an observational study, the assessment being composed of a total of 22 items. This procedure would not be a condition for the study to be included, but rather to identify the ones in which poor-quality assessment could interfere with the outcomes.

## 3. Results

For this systematic review, 41 articles with observational designs were selected. Table 1 shows the main outcomes of each one that will be analysed below.

Table 2 shows the detailed quality assessment of the studies included according to STROBE checklist’s items.

Analysing Figure 2, it can be seen that the vast majority of the studies selected were published in the last five years, suggesting that this topic is still exploratory.

It is assumed that the core aim of them included is to investigate teachers’ visual behaviour in classroom settings. Nevertheless, the independent variables observed and analysed encompass a significant diversity.

For example, several studies ([16]; [44]) compare the visual strategies in terms of teaching experience (between novice and experienced teachers or trainers and teachers in training), other focus on the relationship between teacher visual attention and students’ characteristics, academic performance or behaviour (ethnicity, special educational needs, hand raising movements, among other factors).

Another example are studies exploring the effect of pedagogical training interventions on teachers’ visual behaviour, as well as the effect of factors such as stress, teaching practices (e.g., type of instruction, classroom and dialogue approaches or teaching settings), or even the video visualization perspective or the use of virtual reality technologies ([26]; [48]).

Some studies ([18]; [24]) include the students’ visual perspective as a way of helping to understand teachers’ visual behaviour, while others focus more on the analyses of eye tracking technology as a tool to help self-reflection, with different types of videos and apparatus, with their different impact on the interpretation of teachers’ visual attention and cognitive engagement.

Regarding this last point, in fact, a wide heterogeneity of methodologies used to evaluate teachers’ visual behaviour was found. Studies employing mobile eye-tracking glasses offer a naturalistic approach by capturing real-time gaze data in authentic classroom environments. These tools allow researchers to examine how teachers allocate visual attention while actively engaging with the authentic class ([9]; [23]).

Additionally, other investigations incorporated screen-based eye trackers in controlled simulations or video analysis to assess visual behaviour retrospectively. These approaches are particularly useful to provide detailed patterns in teachers’ gaze behaviour, not invalidating the ecological validity of the studies ([5]; [15]).

The eye-tracking technology research analysed provided various metrics (e.g., fixation counts, durations and time to first fixation), which are all used to determine teacher gaze behaviour. Mobile eye-tracking is particularly useful for capturing teachers’ in-action gaze patterns. Video analysis is also a useful way to study teachers’ professional vision, often combined with eye-tracking data. Additionally, the use of standardised classroom simulations helps to control variations and make comparisons more meaningful ([7]; [45]).

The studies ([19]; [49]) predominantly use quantitative methods and all, as an inclusion criterion, include eye-tracking observation. However, the methodological approaches, as mentioned, varies between real classrooms and in simulations (video analysis with or without teacher commentary).

Other measurements include gathering data on teaching conceptions, stress, attitudes, self-efficacy and other relevant factors, interviews to collect qualitative data on teachers’ perceptions and reflections, post think-aloud analysis where participants verbalize their thoughts while performing a task or tests to evaluate teachers’ pedagogical-psychological knowledge ([12]).

The studies contained encompass a wide variety of educational levels, from early childhood to university context [nursery (*n* = 1), primary school (*n* = 10), middle school (*n* = 12), high school (*n* = 10) and university (*n* = 5)], as well as the subjects (e.g., Mathematics, Science, Geography or History).

The main results and outcomes, which will be developed later in this papers’ Discussion, reveal some consistent patterns. In terms of professional experience, older teachers use more dynamic visual strategies, with more frequent revisits and shorter fixations and focus more quickly on students off-task, while novice tend to concentrate on superficial aspects ([36]).

As expected, other important insights reveal that teachers’ visual behaviour is associated with or influenced by some variables, such as students’ academic skills or profiles of commitment to tasks, teacher stress (negatively affects their visual attention to students) and teaching approach (student-centred practices improve the distribution of this attention) ([5]).

Other studies ([13]; [14]) indicated that teachers’ visual attention varies with their intentions and support strategies and can affect students’ engagement, motivation and satisfaction and that training interventions can improve teachers’ ability to identify and respond to classroom management events. In a direct way, visual attention seems to affect the quality of teaching and learning.

Studies in general (e.g., [13]; [43]; [48]) highlight the importance of eye tracking as a tool for investigating teachers’ visual behaviour and the development of professional vision the influence of experience. Pedagogical training, the need to consider teachers’ didactics filiation and individual differences and the triangulation of appropriate metrics are also mentioned as crucial to understanding the underlying reasoning and developing teachers’ visual attention.

The potential publication bias, where studies reporting significant findings may be more likely to be published than those with inconclusive results, must be considered. Another important factor is the variation in sample populations across studies, such as differences in participants’ teaching experience or subject expertise, aspects that can influence how visual attention is allocated in classroom settings and may limit the generalisability of findings to the teaching context. Nevertheless, this summary of results establishes a solid base for further analysis and for the discussion of the topic of teachers’ professional vision and the importance of eye tracking in research on teaching and learning.

## 4. Discussion

The integration of eye-tracking technology into educational research has provided unprecedented insights into the complexities of teachers’ visual behaviour within classroom settings.

This systematic review of the literature reveals consistent patterns and variations in how teachers allocate their visual attention, which are closely linked to their experience, training and pedagogical approaches. This section delves into the main findings, exploring the differences between experienced and novice teachers, the influence of various factors on gaze patterns and the implications for teacher training and classroom management.

### 4.1. Teaching Experience

Some studies have investigated how teachers’ visual behaviour relates to student performance, often comparing teachers with varying levels of experience ([23]; [35]). The research consistently shows that experienced teachers exhibit different visual strategies compared to novice or pre-service teachers, which can relate to their competence in assessing students and managing classrooms ([29]; [47]).

In relation to their scanpaths and gaze patterns, expert teachers tend to have more complex and recurring ones, often monitoring multiple students more regularly ([22]). They distribute their gaze more evenly across the classroom and return to previously observed students ([12]), suggesting a consistent and student-oriented approach.

With regard to fixation duration and frequency, although there are some contradictory findings, some studies suggest experienced teachers have shorter fixation durations and more frequent fixations on relevant areas ([21]). This indicates that experts can process visual information more quickly and efficiently than novices ([39]). However, other studies have not found significant differences in the number of fixations across professional experience levels ([3]; [17]).

It is also revealed that expert teachers exhibit more complex visual behaviour, which involves monitoring each student more frequently and shifting gaze between all possible student combinations ([22]). All these statements have an impact on assessment competence and students’ and classroom events’ management.

In terms of the attention given to students, experienced teachers demonstrate a student-centred gaze, focusing more on students and their engagement ([32]). They are more likely to visually attend to students who require additional support or who exhibit specific behaviours ([29]).

Expert teachers also are more proficient at noticing relevant classroom events, such as student misbehaviour or critical incidents, and are better at separating task-relevant information from task-redundant information. They can identify and respond more quickly to problematic behaviours ([9]).

The assessment accuracy of students’ characteristics by teachers with more complex gaze patterns and more equal monitoring of students tends to be better. This suggests that the visual strategies employed by expert teachers support their ability to assess students effectively ([21]).

Finally, considering classroom management, experienced teachers can better manage classroom dynamics through visual attention, proactively scanning the classroom and responding to student behaviour ([22]). They effectively distribute their visual attention across all students. Some studies also indicate that experienced teachers are also more able to refocus their attention on relevant aspects after a distraction ([9]).

### 4.2. Trainee and Novice Teachers

Other studies analyse exclusively the visual behaviour of trainee or early career teachers, revealing significant insights into how they visually attend to classroom events ([10]; [34]). Novice teachers display higher variance in the frequency and duration of their eye movements compared to experienced teachers. This is in line with the struggle to focus their attention on information relevant to learning processes, frequently fixing themselves on non-essential aspects of classroom management, due to the difficulty in distributing their attention among all the students in accordance with the requirements of effective teaching and learning ([42]).

Therefore, novice teachers may focus their undivided attention on particular students or instructional materials. They may also be driven by salient features in student behaviour rather than an intention to diagnose students’ cognitive processes. This can result in missing critical events or students who require special attention ([11]).

When analysing teacher-student interactions, it was found that novice teachers are more likely to focus on students exhibiting active, engaged behaviours. Conversely, they might miss students with less obvious cues of engagement, such as the ones who are struggling, underestimating their abilities, or not interested in the topic ([11]).

### 4.3. Other Contextual Variables

There are other specific factors analysed in some of the articles included in relation to visual behaviour, such as, for example, the teaching practices or the students’ characteristics and profile ([4]; [38]).

Teachers do not distribute their visual attention evenly among all students. In this sense, students who actively participate verbally in classes’ activities tend to receive more visual attention from teachers compared to silent students. However, in high-quality educational dialogues, teachers tend to distribute their attention more broadly across students, indicating that effective teaching involves a more inclusive visual focus ([28]).

Expert teachers also tend to have a student-centred gaze, focusing on areas of the classroom that are rich in information, observing the entire classroom and visually attending to students more than to other objects and automatically monitor classroom activity in terms of student engagement and learning ([40]).

About the student behaviour, teachers tend to pay more attention to disruptive or off-task behaviour ([21]). However, when considering self-regulated learning, they seem to focus more on salient behaviours, such as searching for information, than on less visible cognitive and metacognitive regulation behaviours ([15]).

Teachers’ pedagogical intentions also guide their visual attention. For example, during affective scaffolding, teachers focus more on student faces when their intentions are to motivate students and reduce frustration. In this sense, teachers also prioritize student presence during instruction-giving in general ([25]).

These results have an impact on students’ outcomes. When teachers show a balance of gazing at students and teaching content, students tend to be more engaged and motivated, leading to higher satisfaction ([50]). Conversely, when teachers focus more on their own teaching and pedagogical aspects, they tend to focus less on individual students ([27]).

The research also underscores the significance of knowledge and training in developing effective visual strategies. It is argued that teachers’ visual perception is largely driven by top-down processes. This means that these professionals’ gaze is guided by their knowledge, experience and cognitive schemas, rather than eye-catching stimuli ([21]).

Experienced teachers develop knowledge-informed cognitive schemas, which allow them to process and prioritise visual information more effectively. Their capacity to notice relevant classroom features is closely linked to their professional knowledge. Novice teachers often lack the knowledge base to guide their visual attention effectively ([41]).

The interpretation of global measures of visual attention may not always remains a complex and not always a straightforward process, therefore event-related noticing underscores the need for specific pedagogic training to ensure robust interpretations of eye-tracking findings in educational research ([33]). Additionally, combining eye-tracking with other variables, such as think-aloud protocols and analysis of student cues, can give a more comprehensive understanding of the cognitive and behavioural activities driving teacher judgements.

### 4.4. Limitations of the Included Studies

While this review provides valuable insights into teachers’ visual behaviour, several limitations must be acknowledged. The heterogeneity of study designs (e.g., sample populations, methodological procedures and subject taught or other contextual factors) makes direct comparisons between findings difficult. This wide variation might impact the validity and reliability of results, limiting generalisability and making it necessary to formulate and interpretate conclusions cautiously ([46]).

Additionally, in order to fully capture the cognitive and pedagogical decision-making processes of teachers, combining eye-tracking with qualitative methods, such as think-aloud protocols or retrospective interviews, could enhance understanding by analysing gaze patterns with teachers’ reflective insights ([3]; [27]).

## 5. Conclusions

This systematic review of studies points to a trend in which analysing teachers’ visual behaviour through eye-tracking technology provides valuable insights into the complexities of classroom interactions.

The research highlights, not only that experienced teachers demonstrate more effective and efficient visual strategies that benefit their classroom management and assessment capabilities, linked to the development of cognitive domain and professional vision, but also that the integration of eye-tracking data with other methods provides a more comprehensive understanding of the relationship between teachers’ visual attention and student outcomes. In this sense, teachers’ gaze patterns are a key aspect of their professional competence.

In terms of practical implications, as eye-tracking research highlights key areas where inexperienced teachers might struggle with, a deeper understanding of their behaviour can lead to more effective teacher training and better classroom environments for students. Methodological decisions must be carefully considered in this field and should include triangulation of methods and measures and an ecological view of the research, in order to promote more effective pedagogic training for teachers, particularly for novice educators.

The year of publication of the vast majority of the selected studies suggests that this topic is still exploratory, showing that it can and should be investigated in depth. Other limitations of this systematic review, such as the wide variation in educational contexts, which may influence how teachers allocate visual attention and interact with students can be overcome with studies bringing together more robust methodologies thus allowing more reliable analyses of the added value of the eye tracking technology.

It is suggested, not only to establish standardised methodologies in the field, carrying out research with similar apparatus and manipulated variables, as being essential to analyse both global and event-related measures of noticing, but also to use larger and more diverse sample sizes to enhance the generalizability of findings and integrate other data sources (e.g., think-aloud protocols), to provide a more comprehensive understanding strengthen the validity of the results.

Looking to future research, there is the need to investigate how pedagogic training interventions that promote classroom management knowledge can positively impact pre-service teachers’ ability to identify relevant classroom events. Also, due to the lack found and although video observations are still valid and have some strengths, as the authenticity of the learning environment may shape visual behaviour, there is a need for more studies on how teachers’ visual attention operates during actual teaching.

Finally, it is important to investigate other potential confounding variables, such as how teacher education can promote the application of classroom management knowledge to improve teachers’ noticing during their own teaching, allowing for adjustments that enhance student interaction and learning outcomes.

## Figures and Tables

**Figure 1 ejihpe-15-00054-f001:**
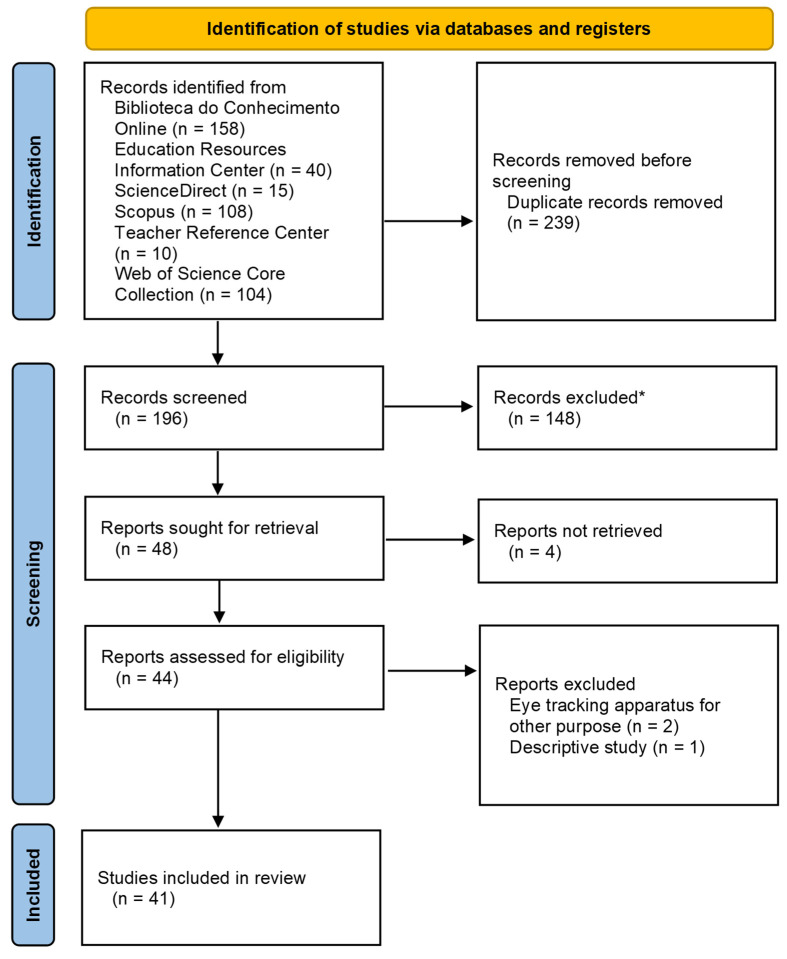
PRISMA 2020 flow diagram for new systematic reviews which included searches of databases and registers only (* Appendix A).

**Figure 2 ejihpe-15-00054-f002:**
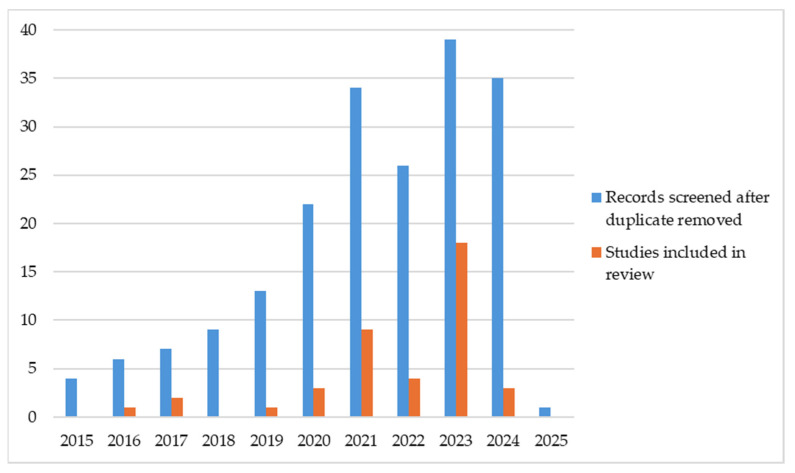
Number of articles included by year of publication.

**Table 1 ejihpe-15-00054-t001:** Main outcomes of the articles included.

Author(s) (Year)	Title	Aim(s)	Participants	Independent Variable(s)	Methodology	Subject(s)	Education Level(s)	Outcomes	Results	Conclusions
[9] ([9])	Eye tracking in a teaching context: comparative study of the professional vision of university supervisor trainers and pre-service teachers in initial training for secondary education in French-speaking Belgium	Compare the visual strategies of university supervisor trainers (UST) and pre-service teachers (PT) when observing a teaching situation	6 UST 16 PT	Teaching experience	Watch a 7′ video in silence with the GazePoint GP3HD Comment by verbalizing their thoughts Share the most salient observations	Geography	10-year-old (primary school)	Fixation time, first view and revisits	UST employed more dynamic visual strategies with more frequent revisits and shorter fixations, focused more quickly on off-task pupils and showed different visual strategies when observing pupils engaged in the lesson PT focused more on actively participating pupil	UST’ visual strategies align with expert teachers, demonstrating a more proactive approach to classroom management, which involves using “glance” strategies to monitor and maintain an overview The study highlights the importance of visual scanning skills in teaching
[15] ([15])	Student self-regulated learning in teacher professional vision: Results from combining result self-reports, teacher ratings, and mobile eye tracking in the high school classroom	Investigate the relationship between student self-regulated learning (SRL), teacher ratings of student SRL and teacher attention distribution during lessons	10 teachers	Student SRL profiles based on student self-reports Teacher ratings of SRL	Teach a lesson (*M* = 39′) wearing the Tobii Pro Glasses 3 Questionnaires (student self-report of practicing SRL and teacher rating of student SRL)	English Mathematics Biology Physics Lithuanian	9th and 10th grades (middle and high school)	Teacher visual attention (number of visits and visit duration)	Student SRL profiles were identified based on the match between teacher and student reports Only one teacher rating scale showed a slight correlation with teacher attention Teachers tend to have a bottom-up approach to student attention in class with regard to SRL, noticing more overt behaviours	SRL is difficult for teachers to assess and their attention to students may be more related to salient student traits and classroom events than to students’ internal SRL processes Triangulation with other data is necessary to understand teacher reasoning behind visual attention
[23] ([23])	Exploring teachers’ eye-tracking data and professional noticing when viewing a 360 video of elementary mathematics	Examine how teachers’ gaze behaviours in an immersive virtual reality-based representation of practice correspond with their professional noticing of children’s mathematics	10 novice preservice teachers 4 experienced teachers	Teaching experience	Watch 6′59″ 360 videos with the Pico Neo 2 Eye VR Describe and select pivotal moments regarding students’ mathematics noticed during the lesson	Mathematics	3rd grade (primary school)	Gaze duration and field of view Levels of specificity in written noticing of student’s mathematics	Experienced teachers had shorter gaze durations and more evenly distributed gaze Teachers with more distributed eye-gaze provided more specific and sophisticated written descriptions of students’ mathematics, focusing on student’s deskwork	Embodied cognition is important to understand professional noticing (experienced teachers focus attention more effectively) There is a need for more research acknowledging physiological data in understanding professional noticing, connecting gaze behaviour with understanding of mathematical reasoning
[3] ([3])	Assessment of noticing of classroom disruptions: a multi-methods approach	Investigate the effectiveness of keystroke and retrospective think-aloud (RTA) methods in assessing the noticing process, triangulated with eye-tracking	26 student teachers 26 experienced teachers	Teaching experience Presence of critical incidents (CI) in videos	Answer knowledge and demographic questions presented online Quasi-randomly watch one video sequence (3′30″) with the Tobii Pro Fusion Answer three questions to explicitly specify the previously identified events	Mathematics Informatics	10th and 11th grades (high school)	Fixation count, mean fixation duration and revisits Accuracy in noticing CI	Participants aware of CI showed higher fixation counts and more revisits to the relevant areas No significant differences were found regarding expertise in the accuracy of noticing CI	Keystroke and RTA methods are promising for assessing noticing, complementing and possibly correcting eye-movement data Methodological triangulation is needed to draw valid conclusions from eye tracking data
[5] ([5])	Teaching practices mediating the effect of teachers’ psychological stress, and not physiological on their visual focus of attention	Examine the associations between teachers’ stress, teaching practices and visual attention and the mediating role of teaching practices	53 teachers	Stress (psychological and physiological–cortisol levels) Teaching practices (child-centred vs teacher-directed)	Give six saliva samples during two working days (cortisol levels) Gerris’s Parental Stress Inventory Early Childhood Classroom Observation Measure from three 45′ lessons Teach 20′/25′ of a lesson wearing Tobii Pro Glasses 2	Literacy Mathematics	1st grade (primary school)	Gini coefficient (distribution of visual attention), total fixation duration and fixation counts on students	Psychological stress indirectly affects visual focus of attention through teaching practices Higher cortisol levels were associated with less visual attention to students More child-centred teaching practices led to more individualised attention distribution and greater number of fixations	Teachers need to develop coping strategies for work-related stress, as it affects their teaching practices and classroom behaviour Child-centred practices improve the distribution of visual attention
[10] ([10])	Guiding pre-service teachers’ visual attention through instructional settings: an eye-tracking study	Examine the effects of specific task instructions and prompts on pre-service teachers’ noticing of classroom management situations	135 pre-service teachers	Type of instruction (specific task instruction, prompting and general task instruction)	Three groups (prompts during 9′18″ videos watching with the Tobii Pro Eye-Tracker Nano, specific task instruction before video and general task instruction before video) Click when identify positive elements of classroom management and those that need improvements Pedagogical-psychological knowledge test	Mathematics German	7th, 8th and 10th grades (middle and high school)	Number of relevant clicks Fixation durations, fixation counts, and gaze relational index	Specific task instructions and prompts led to a higher number of relevant clicks compared to general instructions There was no significant difference on eye-tracking parameters between the three conditions	Task-specific instructions and prompts effectively direct attention to relevant classroom management situations but do not affect the gaze parameters measured
[14] ([14])	University teachers’ professional vision with respect to their conceptions of teaching and learning: findings from an eye-tracking study	Explore the relationship between university teachers’ (mis)conceptions of teaching and learning and their professional vision, and the impact of pedagogical training	32 university teachers	(Mis)conceptions of teaching and learning Pedagogical training (pre- and post-training)	Teachers’ (mis)conceptions questionnaire Watch a 12′ video with the Tobii Pro Spectrum Press down the left mouse button when noticed something pedagogically significant and/or relevant in terms of teaching and learning Stimulated retrospective recall interview		University	Professional vision scores (noticing and interpreting skills) Focus of attention on teacher and student actions (Mis)conceptions of teaching and learning	There was no significant general correlation between professional vision scores and (mis)conceptions of teaching Teachers with less sophisticated conceptions focused on teacher actions Pedagogical training improved the sophistication of teaching conceptions and noticing skills	Pedagogical training is important for university teachers to develop conceptual understanding and professional vision, especially in situations where selective attention is needed
[16] ([16])	Body in motion, attention in focus: A virtual reality study on teachers’ movement patterns and noticing	Identify patterns in pre-service teachers’ movements in an immersive virtual reality (IVR) classroom and their relationship with visual attention performance	21 pre-service teachers	Movement patterns in the IVR classroom	Teach a 4′ lecture in the IVR classroom wearing the HTC Vive Pro Eye system with Tobii XR Teacher’s Sense of Efficacy scale	COVID-19 vaccinations		Number, speed and duration of fixations on student disruptions	Three different movement patterns of preservice teachers were identified Teachers with one of the identified patterns showed significantly less fixations and shorter time to first fixation on student disruptions	Movement patterns of pre-service teachers impact visual attention to student disruptions in an IVR classroom, highlighting the relationship between embodied action and visual focus
[17] ([17])	Classroom chronicles: through the eyeglasses of teachers at varying experience levels	Assess how teaching experience levels influence teachers’ visual processing efficiency, visual span and mental effort during instruction in real classroom settings	22 pre-service teachers 17 beginning teachers 19 experienced teachers	Teaching experience	Teach a lesson (*M* = 44′21″) wearing the SMI 60 Hz glasses Indicate with an inconspicuous hand gesture when experience a remarkable classroom management event during teaching Interview based on the recorded videos	Geography History English Mathematics		Count of fixations, average fixation duration and average fixation dispersion	No significant differences were found for fixation counts across experience levels By the end of the lesson, pre-service teachers showed slightly higher fixation counts and decreased fixation durations Experienced teachers showed a wider visual span at the start of the lesson compared to the end	Professional vision manifests differently across teaching experience levels Pre-service teachers show changes in visual processing by the end of the lesson
[18] ([18])	Mobile eye tracking evoked teacher self-reflection about teaching practices and behavior towards students in higher education	Investigate how mobile eye tracking can contribute to university teachers’ self-reflection and the development of their professional vision	4 university teachers	Use of mobile eye-tracking during lessons Subsequent reflections	Teach a 10′ lecture in the frontal teaching format wearing the Tobii Pro Glasses 3 Two self-reflections watching the video (comment the actions and gaze behaviour during teaching and answer questions aimed at identifying gaps in practice and suggestions for pedagogical alternatives)	Education	University	Number and duration of fixations on students, teacher material, board and other areas Reflection level (descriptive, dialogic and critical)	Teachers prioritized students in lectures, with varying levels of awareness and reflection on their gaze Eye-tracking recording can help teachers reflect critically on their behaviours in the classroom	Mobile eye tracking, combined with reflection, can be a useful tool for teachers’ professional development, promoting awareness of gaze patterns and teaching behaviours
[20] ([20])	Relations between pre-service teacher gaze, teacher attitude, and student ethnicity	Examine differences in pre-service teachers’ fixations on ethnic minority and ethnic majority students and associations with attitudes, self-efficacy and stereotypes	83 pre-service teachers	Ethnicity of students (minority and majority) Teacher attitudes, self-efficacy and stereotypes	Watch a 10′ video with the Tobii Pro Spectrum Answer two questions about the interaction of the teacher with the students Multi-item questionnaire (demographic information and explicit attitudes toward, self-efficacy for teaching and stereotypes about the school-related motivation of ethnic minority students)			Number of fixations, duration of fixations, time to first fixation on minority and majority students Teacher attitudes, self-efficacy and stereotypes	Pre-service teachers fixated longer on ethnic minority students, which is correlated to their positive attitudes towards them No significant differences were found for fixation number or time to first fixation	Teachers’ positive attitudes toward ethnic minority students can influence their visual attention Further research is needed on teacher professional vision and teacher attitudes in diverse classroom contexts
[21] ([21])	Keeping track in classroom discourse: Comparing in-service and preservice teachers’ visual attention to students’ hand-raising behavior	Investigate the relationship between pre-service and in-service teachers’ attentional processes and students’ hand-raising behaviour	10 pre-service teachers 10 in-service teachers	Teaching experience Number of student hand raises	Watch a 7′49″ video with the SMI RED 500	Mathematics	8th grade (middle school)	Number of fixations and gaze patterns	Teacher attentional processes were positively related to the number of hand-raisings In-service teachers’ gazes were more student-centred and distributed attention among more students, regardless of hand-raising behaviour	The study supports the idea that experienced teachers have different cognitive models Eye-tracking can help teachers become aware of their attention distribution and improve their professional development
[27] ([27])	Professional vision in the classroom: Teachers’ knowledge-based reasoning explaining their visual focus of attention to students	Examine the extent to which teachers’ knowledge-based reasoning explains their visual focus of attention to the whole class and individual students	50 teachers	Teachers’ knowledge-based reasoning (description, explanation, prediction) Teaching experience)	Teach a 20′ lesson wearing the Tobii Pro Glasses 2 Retrospective think-aloud interview (recall what they were thinking during their teaching and the reasons for their actions) Questionnaire (teaching experience and the number of students in the classroom)		2nd grade (primary school)	Visual focus of attention (fixation duration and count) Teachers’ knowledge-based reasoning	Teachers’ descriptions of social relations and emotions were positively associated with visual focus of attention to the whole class Teachers’ descriptions of teacher-related information and pedagogy were negatively linked to visual attention to individual students	Teachers’ visual attention varies depending on what guides their attention, such as social relations or pedagogy Teachers should be aware of factors that guide their visual focus of attention There should be more training in teachers’ professional vision
[29] ([29])	University teachers’ focus on students: Examining the relationships between visual attention, conceptions of teaching and pedagogical training	Examine whether university teachers’ focus on students can be observed at the visual level and whether pedagogical training and teaching experience affect visual attention and teaching conceptions	23 novice teachers 25 more experienced teachers	Pedagogical training (trained and untrained) Teaching experience Conception of teaching (learning facilitation and knowledge transmission)	Watch two videos (*M* = 1′35″) with the Tobii TX300 1. Rate the content-focused video from the viewpoint of teaching and learning 2. Think aloud about the learning-focused video Background questionnaire		University	Percentage of fixation time on different areas of interest (students, teacher and slides) Verbal interpretations of the videos	Pedagogically trained teachers paid more visual attention to students in a situation where students were bored Teachers who paid more visual attention to important events also gave more accurate verbal interpretations	Visual attention combined with verbal interpretations can help understand teachers’ pedagogical expertise Pedagogical training may influence the visual attention given to students in the classroom
[32] ([32])	Experienced nursery teachers gaze longer at children during play than do novice teachers: an eye-tracking study	Investigate how experience level affects early childhood education teachers’ visual attention and verbal comments when supervising children at play	10 novice teachers 10 relatively experienced teachers	Teaching experience	Supervise a 10′ class wearing the Tobii Pro Glasses 2 Visual attention interview		3-year-old (nursery)	Gaze patterns on children’s faces and bodies and around play areas Number and type of verbal comments	Experienced teachers spent significantly less time looking at spaces without children The mean gaze duration of experienced teachers was also shorter when looking at the children	Experienced teachers’ attention is more focused on children and less on the surrounding environment
[41] ([41])	The subject matters for the professional vision of classroom management: an exploratory study with biology and mathematics expert teachers	Investigate whether expert teachers from different subjects differ in their professional vision of classroom management (TPVCM)	9 Biology teachers 11 Mathematics teachers	School subject	Watch two videos with the SMI RED-m and press a button whenever noticed an important classroom management event Watch thinking aloud about the events noticed	Biology Mathematics	Middle school	Visual attention Noticing of classroom management events and knowledge-based reasoning about these events	Biology teachers focused more on alternative management strategies and the context of classroom management events Mathematics teachers were more evaluative and focused on behavioural management and ensuring student engagement	Subject-specific aspects of teachers’ professional vision should be considered in teacher education and research The TPVCM may be subject-specific
[43] ([43])	Training & prompting pre-service teachers’ noticing in a standardized classroom simulation—a mobile eye-tracking study	Investigate how training and prompting affect pre-service teachers’ (PST) noticing of classroom management events during a simulation	46 pre-service teachers	Classroom management training and prompting	Three groups (control, training group and training and prompting) Pedagogical Instructional Knowledge (structuring lessons, motivation, dealing with heterogeneity, classroom management and assessing performance) Watch a 20′ video with the Tobii Glasses Pro 3 Retrospective commentaries and observations of their gaze after Assess teachers’ event-related noticing	Sustainability	10th grade (high school)	Fixation count, visit count, average fixation duration, average visit duration and Gini-coefficient Event-related and global noticing of classroom management event	Training and prompting significantly improved event-related noticing, with experimental groups making less time and target errors No significant differences were found in global noticing (fixation counts or duration on students) A correlation between higher noticing accuracy and fixations on students was found	Training and prompting can improve PST’ ability to identify and respond to classroom management events more accurately Knowledge about classroom management is relevant for teachers’ noticing Global measures may be less informative than event-related measures
[44] ([44])	Transitions from presence, belonging to engaged participation in an inclusive classroom: an eye tracking study	Examine how teachers support students with special education needs (SEN) in an inclusive classroom to transit between presence, belonging and engaged participation	1 beginning teacher	Teaching practices	Self-efficacy on teaching SEN student questionnaire Teach a lesson (*M* = 45′) wearing the DIKABLIS Professional Glasses Eye Tracker DG3 Interview (views and experiences thereafter)	Science	4th grade (primary school)	Fixations Transitions between presence, belonging and engaged participation of students with SEN	Transitions between presence, belonging and engagement were not unidirectional and could be truncated by changes in teacher’s eye gaze These transitions were facilitated by teaching practices	Effective inclusive teaching supports transitions from presence to belonging and engaged participation Teachers should be aware of how eye gaze impacts these transitions
[45] ([45])	Teaching on Zoom in the eyes of the lecturer: an eye tracking study	Examine lecturers’ eye gaze patterns while teaching on “Zoom” in higher education	10 lecturers	Online teaching via Zoom	Teach a lesson on “Zoom” wearing the Tobii Pro-Lab 1200 hz Interview to understand the perspective of the lecture			Lecturers’ gaze patterns and perceptions	Lecturers’ fixations were mostly on their presentation slides and then on students Lecturers felt a high interaction with students, although fixations suggested otherwise There was variability in the lecturers’ gaze patterns (they tended to look at themselves to verify they were visible)	The study highlighted the importance of understanding lecturers’ professional vision on “Zoom” to improve the quality of teaching
[48] ([48])	Pre-service and in-service teachers’ professional vision depending on the video perspective–What teacher gaze and verbal reports can tell us	Examine differences in professional vision between pre-service teachers (PT) and in-service teachers (IT) and the influence of video perspectives on their noticing and reasoning	9 pre-service teachers 9 in-service teachers	Teaching experience Video perspective (front, back and eye-tracking)	Watch a 1′30″ video from one of three perspectives (back, front and teacher wearing Tobii Pro Glasses 2) with the Tobii Pro Nano Report what noticed interview	German		Visual fixations Differences between PT and IT noticing and reasoning Influence of video perspective on PT and IT observations	IT described classroom aspects in more detail and explanations than PT Depending on the video perspective, participants focused on different subjects, but the verbal data did not reflect these differences Gaze behaviour and verbal statements were not consistent	Considering multiple sources of data is beneficial to explore professional vision Further research is needed to understand the concept Experienced teachers can grasp situations relevant to learning better
[50] ([50])	The teacher’s eye gaze in university classrooms: Evidence from a field study	Assess experienced university teachers’ eye gaze patterns in classrooms and the impact on students’ engagement, motivation and satisfaction	11 university teachers	Teachers’ eye gaze patterns (student-centred and balanced)	Teach a 10′ lesson wearing the Tobii Pro Glasses 2 Engagement, Motivation and Satisfaction Scales filled out by the students		University	Total fixation duration and count Impact of teachers’ gaze patterns on students’ engagement, motivation, and satisfaction	Teachers with a balanced eye gaze on students and teaching content had students who were more engaged, motivated and satisfied Engagement and motivation mediated the association between teachers’ eye gaze and students’ satisfaction	Allocating attention to both students and teaching content enhances students’ learning experience
[4] ([4])	Teachers’ visual focus of attention in relation to students’ basic academic skills and teachers’ individual support for students: An eye-tracking study	Investigate teachers’ visual focus of attention in relation to students’ basic academic skills and teachers’ individual support for students	46 teachers	Students’ academic skills Teachers’ individual support for students	Teachers rated individual support for students (literacy and math) Teach a 20′–25′ lesson wearing the Tobii Pro Glasses 2 Students’ basic academic skills (Literacy assessment material for 1st grade and Basic Arithmetic Test)	Literacy Mathematics	1st grade (primary school)	Visual focus of attention (fixation counts, total fixation duration and average fixation duration)	Teachers’ fixation counts negatively correlated with students’ basic academic skills and positively with teachers’ individual support for students Average fixation duration was significantly different between classrooms with high and low teacher individual support	Teachers’ visual focus of attention is associated with students’ academic skills and the individual support they receive Teachers’ fixation counts are positively associated with individual support and negatively with academic skills
[12] ([12])	Professional Vision and the Compensatory Effect of a Minimal Instructional Intervention: A Quasi-Experimental Eye-Tracking Study With Novice and Expert Teachers	Investigate how prospective and experienced teachers perceive teaching situations and whether instructional support changes gaze behaviour	34 pre-service teachers 37 experienced teachers	Teacher experience	Answer demographic data and the Pedagogical/Psychological Knowledge Test Quasi-randomly watch one video sequence (*M* = 1′43″) with the Tobii Pro Fusion Identify relevant events via keyboard-press Answer three questions to explicitly specify the previously identified events	Mathematics Informatics	10th and 11th grades (high school)	Global monitoring gaze and event-related gaze (fixation count, visit count, mean fixation duration, visit duration and gaze relational index)	No expertise-dependent differences were found in eye-tracking parameters Minimal intervention did not improve professional vision for experts or novices	Expertise effects were not replicated in the experiment, but specific instruction can influence gaze movement in terms of fixation count
[19] ([19])	Noticing and weighing alternatives in the refection of regular classroom teaching: Evidence of expertise using mobile eye-tracking	Examine differences in teachers’ professional vision when reflecting on their own teaching videos	37 teachers	Teaching experience	Teach a lesson wearing the ASL Mobile Eye Tracker Freely comment the video	Mathematics English Social studies History	1st to 11th grades (primary, middle and high school)	Number of classroom events noticed Perception of noticed events (negative, positive, or neutral) Number of alternative teaching strategies mentioned Reasons for mentioning alternative teaching strategies (negative, positive or neutral	Did not differ significantly in the number of classroom events noticed or alternative teaching strategies mentioned Novice teachers were more critical of their own teaching, noticing more negative events than positive, whereas expert teachers more positive Novice teachers suggested alternative strategies to improve their teaching more often	Mobile eye-tracking video feedback can be a useful tool for both expert and novice teachers to enhance their reflection on teaching practices It helps teachers notice events they may have missed during the lesson and consider alternative strategies Novice teachers may need additional support to balance their negative self-evaluations
[39] ([39])	Through Teachers’ Eyes: An Eye-Tracking Study of Classroom Interactions	Analyse vision data from pre-service and in-service teachers to understand the development of professional vision across career levels	43 junior pre-service teachers 46 senior pre-service teachers 17 experienced in-service teachers	Teacher experience	Watch four video sequences (15′15″) with the Tobii X2-60 Provide written comments on what they saw		11 to 13-year-old (middle school)	Fixation duration, fixation count and time to first fixation on areas of interest	Significant differences were found in fixation duration and fixation count, especially in semi-structured situations Less experienced teachers fixed earlier on peripheral areas compared to in-service teachers	Visual processing develops across career levels, with pre-service teachers being more sensitive to classroom dynamics and misbehaviour
[7] ([7])	Investigation of classroom management skills by using eye-tracking technology	Investigate the effect of wearable eye-tracking technology on the enhancement of instructors’ visual behaviours concerning classroom management	2 instructors	Use of wearable eye-tracking technology and video camera recordings	Teach three lessons (*M* = 45′) wearing the Tobii Pro Glasses 2100 Hz Live View Wireless Comment the videos on their classroom management performance	Education Physics	University	Visual attention patterns, eye contact and interactions with classroom technology	Eye-tracking provides valuable feedback on instructors’ visual attention and how they interact with classroom technology Instructors use eye contact to manage student behaviour and tend to focus on the students perceived to be misbehaving Instructors’ physical actions can hinder eye contact with students	Wearable eye-tracking technology, combined with video analysis and Retrospective Think aloud sessions is a useful tool for improving classroom management skills It can be used to train instructors, especially on non-verbal communication and to evaluate instructors’ interactions with technology
[11] ([11])	How does learners’ behavior attract preservice teachers’ attention during teaching?	Explore how learners’ behaviour affects pre-service teachers’ attention and if this changes over time	4 pre-service teachers	Learner behaviour (uninterested, under-estimating, struggling or strong profiles)	Teach a lesson (*M* = 15′19″) wearing eye-tracking glasses Students’ behaviour and teachers’ instructional practices rating			Attentional focus per area of interest	The effect of learners’ behaviour on pre-service teachers’ attention does change over time and there are profile specific differences	Pre-service teachers’ attentional focus is influenced by learner behaviour, with variations over time and depending on learner profiles
[22] ([22])	Identifying Expert and Novice Visual Scanpath Patterns and Their Relationship to Assessing Learning-Relevant Student Characteristics	Explore how eye-movement patterns (scanpaths) differ across expert and novice teachers during an assessment situation	35 novice teachers 9 in-service teachers	Teacher experience	Watch 11′ two videos segment (whole-class instruction and individual work) with the SMI RED 500 Assign five marked students to one of the five listed student profiles (strong, struggling, overestimating, underestimating and uninterested)	Mathematics	8th grade (middle school)	Number of scanpaths and visual transitions	Expert teachers’ scanpaths were more complex, covering more students, indicating a strategy of monitoring all students more equally Novice teachers tended to focus on fewer students, with recurring transitions between just two students Experts also demonstrated higher accuracy in assessing students	Expert teachers employ more complex visual strategies that lead to more accurate assessments of student characteristics compared to novice teachers
[25] ([25])	Students in sight: Using mobile eye-tracking to investigate mathematics teachers’ gaze behaviour during task instruction giving	Investigate how instructional types influence teachers’ gaze for student presence and engagement	6 teachers	Instructional type (introductory, collaborative and reflective)	Teach a 45′ lesson wearing eye-tracking glasses Teachers delivered three task instruction types (introductory, collaborative and reflective) to be along with problem-solving	Mathematics	9th grade (middle school)	Proportional and mean dwell durations and scanpaths Verbalizations	Teachers prioritised student presence across all types of instruction, with no significant differences between instruction types, although they did tend to look more at student faces and bodies, particularly at the start of a lesson	Teachers prioritize student presence and engagement regardless of instruction type
[26] ([26])	Teachers’ Professional Vision: Teachers’ Gaze During the Act of Teaching and After the Event	Compare teachers’ gaze during the act of teaching (IN mode) and after the event (ON mode)	3 teachers	Observation mode (in-action and on-reflection)	Teach four lessons (*M* = 45′) wearing the SMI Eye Tracking Glasses 2 Wireless Watch four 1′–2′ video sequence with the SMI RED250 Comment the video sequences	English	4th, 6th and 7th grades (primary and middle school)	Dwell times	The greatest differences in attention to individual pupils occurred when a pupil who was interacted with during the situation was missing from the video recording In the ON mode, teachers monitored more pupils more often	Teachers’ gaze behaviour varies between in-action and reflection, influenced by the field of view Monitoring is more focused in action and more comprehensive when reflecting
[35] ([35])	Student Characteristics in the Eyes of Teachers: Differences Between Novice and Expert Teachers in Judgment Accuracy, Observed Behavioral Cues, and Gaze	Investigate differences in teacher diagnostic skills between novice and expert teachers when observing student engagement and inferring underlying student characteristic profiles	27 novice teachers 7 expert teachers	Teacher experience Student profiles (strong, struggling, overestimating, underestimating and uninterested)	Watch 7′30″ two videos segment (whole-class instruction and individual work) with the SMI RED 500 Assign five marked students to one of the five listed student profiles	Geometry	8th grade (middle school)	Judgement accuracy Observed student cues Gaze (fixation count and duration)	Expert teachers were more accurate in judging incoherent profiles (overestimating, underestimating, and uninterested), stated more indicators for overestimating and uninterested students and spent more time looking at students who might need support during seatwork	Teacher gaze can serve as an additional operationalisation of the noticing component of teacher professional vision Expert teachers tend to pay more attention to students who might need further support
[36] ([36])	Teachers’ visual processing of children’s offtask behaviors in class: A comparison between teachers and student teachers	Compare the visual processing of off-task behaviours in class between teachers and student teachers	76 in-service teachers 147 student teachers	Teacher experience	Watch a 50″ video with the T60 Questionnaire about children warned		3rd grade (primary school)	Number of fixations, fixation duration and fixation duration per fixation	Teachers had a significantly higher number of fixations on the target children’s areas of interest AOI in the second half of the video and showed a more frequent gaze toward the target child	Experienced teachers gaze more frequently at areas where off-task behaviour is occurring Teachers use their working memory efficiently, successfully performing various complex cognitive activities in the classroom
[40] ([40])	Novice and expert teachers’ noticing of classroom management in whole-group and partner work activities: Evidence from teachers’ gaze and identification of events	Investigate how novice and expert teachers’ noticing of classroom management (CM) events differs regarding whole-group instruction and partner work	40 pre-service teachers 40 in-service teachers	Teacher experience Instructional format (whole-group and partner work)	Watch four videos (1′–2′) twice with the SMI RED-m 1. Push a button every time consider a CM relevant 2. Report CM noticed	Biology Mathematics	Middle school	Number of CM events noticed Visual attention (proportion of gaze and number of fixations) to student groups and the teacher, and specific CM events	In the whole-group format, experts showed a higher proportion of gaze towards the left student group (same and right student group in the partner format) and novices had more fixations on the teacher (same in the partner format)	Expertise influences visual attention to students but not to specific CM events Findings should not be generalised across different formats of instruction
[49] ([49])	Student Teachers’ and Teacher Educators’ Professional Vision: Findings from an Eye Tracking Study	Explore professional vision in student teachers and teacher educators using eye tracking and post hoc think-aloud verbalisations	28 student teachers 28 teacher educators	Teacher experience	Watch a 50′ video with the GazePoint GP3HD Desktop Comment the video		6 to 10-year-old (primary school)	Fixation counts and aggregated total fixation duration on areas of interest Verbalisations of critical incident	Six teacher educators explicitly mentioned the critical incident in post-hoc think-aloud verbalisation, whereas none of the student teachers did and showed more fixations on the student involved in the critical incident and less on the teacher	Eye-tracking data can assist in identifying professional vision, primarily regarding the “noticing” component, when combined with content-related verbalisation
[28] ([28])	Dialogue through the eyes: Exploring teachers’ focus of attention during educational dialogue	Explore teachers’ focus of attention during educational dialogue and whether it varies with the quality of the dialogue	51 teachers	Quality of educational dialogue (moderate and high)	Teach a 20′ lesson wearing the Tobii Pro Glasses 2 Analysis of five educational dialogue principles (collectivity, reciprocity, supportiveness, cumulativity and purposefulness) and two episodes (moderate and high-quality)	Literacy Mathematics Science Art	1st grade (primary school)	Gaze distribution (average fixation count and duration) Verbal participation of students Quality of educational dialogue	Teachers distribute visual attention unevenly during educational dialogue, with more attention given to verbally participating students More students received visual attention during high-quality dialogue	Teachers monitor their classrooms during discussions and invite students into discussion with their focus of attention (high-quality discussions include more students receiving visual attention)
[34] ([34])	Connecting Judgment Process and Accuracy of Student Teachers: Differences in Observation and Student Engagement Cues to Assess Student Characteristics	Connect teachers’ judgment accuracy to judgment processes, specifically looking at eye movements and use of student cues	43 student teachers	Level of judgment accuracy (high and low)	Watch 11′ two videos segment (individual work and whole class) with the SMI RED 500 Assign five marked students to one of the five listed student profiles Assess five student cue utilization (behavioural, cognitive, emotional engagement, knowledge and student confidence)	Mathematics	8th grade (middle school)	Judgment accuracy Eye movement patterns (fixation count and average fixation duration) Utilization of student cues	Teachers with high accuracy showed eye movement patterns similar to experienced teachers (more fixations and shorter durations) and used specific combinations of student cues more consistently Teachers with low accuracy tended to use many different cue co-occurrences, including misleading combinations	Accurate judgments are related to the use of specific student cues and an “experienced” eye movement pattern, suggesting a knowledge-driven process Teacher education should support student teachers in observing and using diagnostic student cues and the development of professional vision
[38] ([38])	Teachers’ gaze over space and time in a real-world classroom	Investigate gaze distribution of experienced teachers in real-world classrooms, examining equality of gaze and its relationship with student characteristics	3 teachers	Student characteristics (gender, achievement level and seating position)	Teach four 45′ lessons wearing the SMI Eye Tracking Glasses 2 Wireless Students’ characteristics (grades high and low) Interview (and teachers’ professional background and impressions of the lessons)	English	5th and 6th grades (middle school)	Gaze distribution (average dwells, number of fixations, relative dwell time and fixation durations) Relationship between gaze and student characteristics	Substantial variation in gaze distribution between teachers and individual lessons was found, as well as towards individual students No effects of student gender were observed Different metrics yielded different perspectives on gaze distribution	The results highlight the importance of considering individual variation in teacher gaze and selecting appropriate metrics for analysis Gaze distribution does not consistently become more equal with experience Teachers’ focuses of attention may not be directly linked with the quality of teaching
[13] ([13])	Teacher’s visual attention when scaffolding collaborative mathematical problem solving	Investigate teacher’s visual attention during scaffolding interactions in a mathematics lesson	1 teacher	Scaffolding interaction categories (cognitive, affective, metacognitive, monitoring and fading)	Teach an 18′ lesson wearing 3D-printed eyeglasses Watch the video and explain actions and thinking Categorize five teachers’ scaffolding interaction (cognitive, affective, metacognitive, monitoring and fading)	Mathematics	15 and 16-year-old (high school)	Number and durations of gazes	Teacher’s scaffolding intentions influenced their visual attention Gaze patterns differed depending on the type of scaffolding For instance, longer gaze durations were observed on student papers during cognitive scaffolding	Intentional attention patterns differ based on scaffolding intentions Gaze data combined with verbal data allows for a detailed analysis of teachers’ attentional behaviours during interactions
[24] ([24])	Visual Attention of Science Class: An Eye-tracking Case Study of Student and Teacher	Identify the characteristics of both students’ and teachers’ visual attention and how they interpret non-verbal and verbal information during a science class	1 student 1 teacher	Role of the participant	Watch a 50′ lesson wearing the Tobii’s Pro Glasses 2 (student) Teach a 50′ lesson wearing the Tobii’s Pro Glasses 2 (teacher) Retrospective interview about the effect of eye-tracking on science class	Science	High school	Total fixation numbers, durations and first fixation time in each area of interest	Students’ visual attention declined 15 min after the start of the class and focused primarily on the teacher’s face and experiment and learning tools Students engaged in more cognitive processing of information from teacher’s face or body language Teachers focused heavily on students in the middle of the classroom	Students and teachers use cognitive effort to interpret nonverbal information Teachers monitor student behaviour and the state of specific students to understand class progress It is suggested that teachers should focus on understanding students’ cognitive states, maintain student attention and set appropriate reference students
[42] ([42])	What is in the eye of preservice teachers while instructing? An eye-tracking study about attention processes in different teaching situations	Analyse study attention processes of pre-service teachers during instruction in different teaching situations	7 preservice teachers	Teaching setting (standardised university setting and real classroom)	Teach two lessons (standardized instructional and real situation) wearing the SMI—SensoMotoric Instruments		High school	Distribution of attention (fixation frequency and duration of total fixation)	Pre-service teachers focus most frequently on simulated learners followed by instructional materials There were no significant differences in focus of attention between the two settings Preservice teachers’ attentional processes are similar to novice teachers	Attention processes during teaching can be studied using eye-tracking Pre-service teachers’ attentional processes are similar when teaching in simulated and real classroom settings
[47] ([47])	Teacher vision: expert and novice teachers’ perception of problematic classroom management scenes	Investigate how teachers’ expertise affects their perception and interpretation of classroom management situations	35 experienced teachers 32 pre-service teachers	Teaching experience	Watch four 2′–4′ two video types (unrelated and interrelated classroom events) with the SMI RED250 Watch again and think aloud what were thought		High school	Distribution of fixations, fixation dispersion average and areas of interest visits and skips Word usage in verbalisations	Expert teachers integrate concerns of teaching and learning, while novices focus on surface-level issues Experts and novices differ in where they focus their attention in classroom videos and what information they ignore	Teachers’ professional vision is linked to their ability to notice and interpret classroom information and can be investigated through eye-tracking and verbal analysis

**Table 2 ejihpe-15-00054-t002:** Quality assessment of the articles included.

Author(s) (Year)	1		2	3	4	5	6	7	8	9	10	11	12					13			14			15	16			17	18	19	20	21	22
	a	b											a	b	c	d	e	a	b	c	a	b	c		a	b	c						
[9] ([9])	✓	✓	✓	✓	✓	✓	✓	✓	✓	✓	X	✓	✓	✓	X	X	n.a.	X	X	X	✓	✓	n.a.	✓	✓	✓	n.a.	n.a.	✓	✓	✓	✓	✓
[15] ([15])	✓	✓	✓	✓	✓	✓	✓	✓	✓	X	X	✓	✓	✓	X	X	X	X	X	X	✓	✓	n.a.	✓	✓	X	n.a.	X	✓	✓	✓	✓	X
[23] ([23])	✓	✓	✓	✓	✓	✓	✓	✓	✓	X	X	✓	✓	✓	X	X	n.a.	X	X	X	✓	✓	n.a.	✓	✓	X	n.a.	n.a.	✓	✓	✓	✓	✓
[3] ([3])	✓	✓	✓	✓	✓	✓	✓	✓	✓	X	X	✓	✓	✓	✓	X	X	X	X	X	✓	✓	n.a.	✓	✓	X	n.a.	X	✓	✓	✓	✓	✓
[5] ([5])	✓	✓	✓	✓	✓	✓	✓	✓	✓	X	X	✓	✓	✓	✓	X	X	X	X	X	✓	✓	n.a.	✓	✓	✓	n.a.	X	✓	✓	✓	✓	✓
[10] ([10])	✓	✓	✓	✓	✓	✓	✓	✓	✓	✓	X	✓	✓	✓	X	X	n.a.	X	X	X	✓	✓	n.a.	✓	✓	X	n.a.	n.a.	✓	✓	✓	✓	✓
[14] ([14])	✓	✓	✓	✓	✓	✓	✓	✓	✓	X	X	✓	✓	✓	✓	X	X	X	X	X	✓	✓	n.a.	✓	✓	X	n.a.	X	✓	✓	✓	✓	✓
[16] ([16])	✓	✓	✓	✓	✓	✓	✓	✓	✓	✓	✓	✓	✓	✓	✓	X	n.a.	X	X	X	✓	✓	n.a.	✓	✓	X	n.a.	n.a.	✓	✓	✓	✓	X
[17] ([17])	✓	✓	✓	✓	✓	✓	✓	✓	✓	✓	✓	✓	✓	✓	✓	X	n.a.	X	X	X	✓	✓	n.a.	✓	✓	X	n.a.	n.a.	✓	✓	✓	✓	✓
[18] ([18])	✓	✓	✓	✓	✓	✓	✓	✓	✓	✓	✓	✓	✓	✓	X	X	X	X	X	X	✓	✓	n.a.	✓	✓	n.a.	n.a.	X	✓	✓	✓	✓	✓
[20] ([20])	✓	✓	✓	✓	✓	✓	✓	✓	✓	X	✓	✓	✓	✓	X	X	X	X	X	X	✓	✓	n.a.	✓	✓	✓	n.a.	X	✓	✓	✓	✓	X
[21] ([21])	✓	✓	✓	✓	✓	✓	✓	✓	✓	X	✓	✓	✓	✓	X	X	X	X	X	X	✓	✓	n.a.	✓	✓	✓	n.a.	X	✓	✓	✓	✓	✓
[27] ([27])	✓	✓	✓	✓	✓	✓	✓	✓	✓	X	✓	✓	✓	✓	X	X	n.a.	X	X	X	✓	✓	n.a.	✓	✓	X	n.a.	n.a.	✓	✓	✓	✓	✓
[29] ([29])	✓	✓	✓	✓	✓	✓	✓	✓	✓	✓	✓	✓	✓	✓	✓	X	X	X	X	X	✓	✓	n.a.	✓	✓	✓	n.a.	X	✓	✓	✓	✓	X
[32] ([32])	✓	✓	✓	✓	✓	✓	✓	✓	✓	✓	✓	✓	✓	✓	X	X	n.a.	X	X	X	✓	✓	n.a.	✓	✓	✓	n.a.	n.a.	✓	✓	✓	✓	✓
[41] ([41])	✓	✓	✓	✓	✓	✓	✓	✓	✓	X	✓	✓	✓	✓	X	X	n.a.	X	X	X	✓	✓	n.a.	✓	✓	n.a.	n.a.	n.a.	✓	✓	✓	✓	✓
[43] ([43])	✓	✓	✓	✓	✓	✓	✓	✓	✓	X	✓	✓	✓	✓	✓	X	n.a.	X	X	X	✓	✓	n.a.	✓	✓	X	n.a.	n.a.	✓	✓	✓	✓	X
[44] ([44])	✓	✓	✓	✓	✓	✓	✓	✓	✓	X	X	✓	✓	✓	X	X	n.a.	X	X	X	✓	✓	n.a.	✓	✓	n.a.	n.a.	n.a.	✓	✓	✓	✓	X
[45] ([45])	✓	✓	✓	✓	✓	✓	✓	✓	✓	X	X	✓	✓	✓	X	X	n.a.	X	X	X	✓	✓	n.a.	✓	✓	n.a.	n.a.	n.a.	✓	✓	✓	✓	✓
[48] ([48])	✓	✓	✓	✓	✓	✓	✓	✓	✓	✓	X	✓	✓	✓	✓	X	X	X	X	X	✓	✓	n.a.	✓	✓	X	n.a.	X	✓	✓	✓	✓	✓
[50] ([50])	✓	✓	✓	✓	✓	✓	✓	✓	✓	X	✓	✓	✓	✓	X	X	n.a.	X	X	X	✓	✓	n.a.	✓	✓	✓	n.a.	n.a.	✓	✓	✓	✓	✓
[4] ([4])	✓	✓	✓	✓	✓	✓	✓	✓	✓	X	X	✓	✓	✓	X	X	X	X	X	X	✓	✓	n.a.	✓	✓	✓	n.a.	X	✓	✓	✓	✓	✓
[12] ([12])	✓	✓	✓	✓	✓	✓	✓	✓	✓	✓	✓	✓	✓	✓	✓	X	n.a.	X	X	X	✓	✓	n.a.	✓	✓	✓	n.a.	n.a.	✓	✓	✓	✓	✓
[19] ([19])	✓	✓	✓	✓	✓	✓	✓	✓	✓	X	X	✓	✓	✓	✓	X	n.a.	X	X	X	✓	✓	n.a.	✓	✓	✓	n.a.	n.a.	✓	✓	✓	✓	✓
[39] ([39])	✓	✓	✓	✓	✓	✓	✓	✓	✓	X	X	✓	✓	✓	✓	X	n.a.	X	X	X	✓	✓	n.a.	✓	✓	X	n.a.	n.a.	✓	✓	✓	✓	X
[7] ([7])	✓	✓	✓	✓	✓	✓	✓	✓	✓	X	X	✓	✓	✓	X	X	n.a.	X	X	X	✓	✓	n.a.	✓	✓	n.a.	n.a.	n.a.	✓	✓	✓	✓	X
[11] ([11])	✓	✓	✓	✓	✓	✓	✓	✓	✓	✓	✓	✓	✓	✓	✓	X	n.a.	X	X	X	✓	✓	n.a.	✓	✓	n.a.	n.a.	n.a.	✓	✓	✓	✓	✓
[22] ([22])	✓	✓	✓	✓	✓	✓	✓	✓	✓	X	✓	✓	✓	✓	X	X	n.a.	X	X	X	✓	✓	n.a.	✓	✓	✓	n.a.	n.a.	✓	✓	✓	✓	✓
[25] ([25])	✓	✓	✓	✓	✓	✓	✓	✓	✓	✓	✓	✓	✓	✓	X	X	n.a.	X	X	X	✓	✓	n.a.	✓	✓	n.a.	n.a.	n.a.	✓	✓	✓	✓	X
[26] ([26])	✓	✓	✓	✓	✓	✓	✓	✓	✓	✓	✓	✓	✓	✓	X	X	n.a.	X	X	X	✓	✓	n.a.	✓	✓	X	n.a.	n.a.	✓	✓	✓	✓	✓
[35] ([35])	✓	✓	✓	✓	✓	✓	✓	✓	✓	X	✓	✓	✓	✓	X	X	X	X	X	X	✓	✓	n.a.	✓	✓	X	n.a.	X	✓	✓	✓	✓	✓
[36] ([36])	✓	✓	✓	✓	✓	✓	✓	✓	✓	X	X	✓	✓	✓	X	X	n.a.	X	X	X	✓	✓	n.a.	✓	✓	X	n.a.	n.a.	✓	✓	✓	✓	✓
[40] ([40])	✓	✓	✓	✓	✓	✓	✓	✓	✓	X	✓	✓	✓	✓	X	X	X	X	X	X	✓	✓	n.a.	✓	✓	✓	n.a.	X	✓	✓	✓	✓	✓
[49] ([49])	✓	✓	✓	✓	✓	✓	✓	✓	✓	✓	✓	✓	✓	✓	✓	X	n.a.	X	X	X	✓	✓	n.a.	✓	✓	X	n.a.	n.a.	✓	✓	✓	✓	X
[28] ([28])	✓	✓	✓	✓	✓	✓	✓	✓	✓	X	✓	✓	✓	✓	X	X	n.a.	X	X	X	✓	✓	n.a.	✓	✓	X	n.a.	n.a.	✓	✓	✓	✓	✓
[34] ([34])	✓	✓	✓	✓	✓	✓	✓	✓	✓	✓	✓	✓	✓	✓	X	X	n.a.	X	X	X	✓	✓	n.a.	✓	✓	✓	n.a.	n.a.	✓	✓	✓	✓	✓
[38] ([38])	✓	✓	✓	✓	✓	✓	✓	✓	✓	✓	✓	✓	✓	✓	X	X	X	X	X	X	✓	✓	n.a.	✓	✓	✓	n.a.	X	✓	✓	✓	✓	✓
[13] ([13])	✓	✓	✓	✓	✓	✓	✓	✓	✓	X	✓	✓	✓	✓	X	X	n.a.	X	X	X	✓	✓	n.a.	✓	✓	n.a.	n.a.	n.a.	✓	✓	✓	✓	✓
[24] ([24])	✓	✓	✓	✓	✓	✓	✓	✓	✓	X	X	✓	✓	✓	X	X	n.a.	X	X	X	✓	✓	n.a.	✓	✓	n.a.	n.a.	n.a.	✓	✓	✓	✓	X
[42] ([42])	✓	✓	✓	✓	✓	✓	✓	✓	✓	X	✓	✓	✓	✓	✓	X	n.a.	X	X	X	✓	✓	n.a.	✓	✓	n.a.	n.a.	n.a.	✓	✓	✓	✓	X
[47] ([47])	✓	✓	✓	✓	✓	✓	✓	✓	✓	X	✓	✓	✓	✓	✓	X	n.a.	X	X	X	✓	✓	n.a.	✓	✓	X	n.a.	n.a.	✓	✓	✓	✓	X

Legend: ✓—fulfils the criteria; X—does not fulfil the criteria; n.a.—the criteria does not apply.

## Data Availability

We will share the research data with MDPI.

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
