# Peer review of "Analysis of Teachers’ Visual Behaviour in Classes: A Systematic Review"

_ejihpe, 2025, doi:10.3390/ejihpe15040054_

Round 1

Reviewer 1 Report

Comments and Suggestions for Authors

Thank you to the authors for a thorough review.

I will list a few minor points here that could improve this publication.

First, I really appreciate that the authors included in the appendix the list of articles that they did not select for this study. This provides transparency and understanding of their choices and bias that may have been introduced. And it is a good tool of interest to researchers in general to have such a comprehensive list of articles.

I also like that the authors used other languages besides English, such as Portuguese, Spanish, and French. I think this broadens the coverage of articles in the field. However, the authors should mention this more explicitly, as this is a disadvantage for articles published in other languages, such as German.

I would like to challenge the authors to think about what eye tracking actually measures and whether we really know. It is not the focus of this paper, but still I would like to recommend the authors to read the work of Schindler, M., & Lilienthal, A.J. (2019) just for their reflection. And I list a second reference that I think is also relevant.

And one last important comment – I think the authors could improve the relevance of their work and help readers to use their work better. I recommend that the authors organize the Discussion section into subsections based on the main themes that emerged from their study, thus facilitating faster access to their findings.

Beach, P. & McConnel, J. (2019). Eye tracking methodology for studying teacher learning: a review of the research. International Journal of Research & Method in Education, 42(5), 485-501. https://doi.org/10.1080/1743727X.2018.1496415 

Schindler, M., & Lilienthal, A.J. (2019). Domain-specific interpretation of eye tracking data: towards a refined use of the eye-mind hypothesis for the field of geometry. Educ Stud Math 101, 123–139. https://doi.org/10.1007/s10649-019-9878-z 

Author Response

First, I really appreciate that the authors included in the appendix the list of articles that they did not select for this study. This provides transparency and understanding of their choices and bias that may have been introduced. And it is a good tool of interest to researchers in general to have such a comprehensive list of articles.

R.: We deeply appreciate the reviewer's comments, which have received our utmost attention. We sincerely thank you for the relevance of your contribution, which will undoubtedly enhance the quality of this article.

I also like that the authors used other languages besides English, such as Portuguese, Spanish, and French. I think this broadens the coverage of articles in the field. However, the authors should mention this more explicitly, as this is a disadvantage for articles published in other languages, such as German.

R.: Thank you for your proposal. We have incorporated it on page 5 (lines 136-140).

I would like to challenge the authors to think about what eye tracking actually measures and whether we really know. It is not the focus of this paper, but still I would like to recommend the authors to read the work of Schindler, M., & Lilienthal, A.J. (2019) just for their reflection. And I list a second reference that I think is also relevant.

R.: Thank you for your proposal. We have incorporated them on page 2 (lines 48-53) and on page 31 (lines 369-376).

And one last important comment – I think the authors could improve the relevance of their work and help readers to use their work better. I recommend that the authors organize the Discussion section into subsections based on the main themes that emerged from their study, thus facilitating faster access to their findings.

R.: Thank you for your proposal. We have incorporated them on pages 29-31.

For reference, please see the following text on page 33 (lines 442-451): “As a systematic review, the Institutional Review Board Statement and Informed Consent Statement do not apply. However, the subsequent experimental scientific studies, which are already underway, have been approved by the Ethics Committee of the University of Coimbra, Faculty of Sports Sciences and Physical Education (reference CE/FCDEF-UC/00142024).

Reviewer 2 Report

Comments and Suggestions for Authors

Dear researchers, I attach a Word document with some comments.

Sincerely,

Author Response

First of all, I would like to highlight several very positive things in your review. The good use of the PRISMA guide, the registration of the review in INPLASY, for including 41 articles in your review and for incorporating Appendix A detailing the reasons for the exclusions. In addition, it is always important to look for improvements in the teaching and learning process.

R.: We deeply appreciate the reviewer's comments, which have received our utmost attention. We sincerely thank you for the relevance of your contribution, which will undoubtedly enhance the quality of this article.

Introduction - There are quite a few statements that could include a quote, to strengthen the importance of the development and justification of this topic. For example: in line 45, "Eye-tracking technology has acquired considerable relevance in educational settings" and the paragraph that continues "By capturing and quantifying eye movements, saccades and fixations, eye-tracking apparatus enables scientific researchers to decode visual attention and cognitive processes". I think that both statements need a citation. The same thing happens to me with the following sentence in line 54 "Teachers' visual behaviour has an important role in learning and instruction". A very powerful statement that needs a citation.

R.: Thank you for your proposal. We have incorporated them on pages 1-2 (lines 45-60).

Although I understand that the use of the word fragmented seeks to give a synonym to the word heterogeneity, I believe that it can be interpreted differently. I recommend looking for a better synonym (line 64-66).

R.: Thank you for your proposal. We have incorporated it on page 2 (line 79) with the word “variety”.

Materials and Methods - Why didn't you use PubMed? It is one of the most widely used metasearch engines for reviews.

R.: Thank you for your comment. Considering the inclusion criteria, no articles eligible for this systematic review were found in PubMed in a previous search.

In the inclusion criteria, are any type of teacher included? That is, it does not matter what level the classes are taught at?

R.: Thank you for your comment. In this case, all levels of education - from nursery to university - were included to expand the scope of this systematic review. This option has been made based concerning the belief that wide range of studies is necessary in this moment in this area of knowledge.

Results - In Figure 2 you could add the exact number of articles found per year, to make it easier for the reader to understand. I think it would be a good idea to publish it as a graph.

R.: Thank you for your proposal. We have incorporated it on page 5.

I have a big doubt about this sentence in the results "For this systematic, 41 articles with observational and experimental designs were selected. Table 1 shows the main outcomes of each one that will be analyzed below." Why do you talk about experimental if the STROBE checklist is only for observational studies? I need to clarify that or correct the concept of experimental that lends itself to confusion. I also recommend changing systematic to review or systematic review, it reads much better.

R.: Thank you for your proposal. We have incorporated it on page 7 (lines 170-171) with the following text: “For this systematic review, 41 articles with observational designs were selected. Table 1 shows the main outcomes of each one that will be analysed below.

Although it perfectly summarizes the wide diversity of results found, I would like to expand a little on the different methodologies and ways of evaluating visual behavior. I recommend going a little deeper into that idea, seeking to show more explicitly the wide heterogeneity found in the results. For example, contrasting the different technological instruments they used to perceive the teachers' visual behavior and specify which ones were used in a real classroom context and which ones were used in simulation. Also, those who performed video analysis.

R.: Thank you for your proposal. We have incorporated it on page 28 (lines 196-208).

Discussion - I congratulate you on the discussion section, I found it to be very well addressed, developed and justified.

R.: We deeply appreciate the reviewer's comments, which have received our utmost attention. We sincerely thank you for the relevance of your contribution, which will undoubtedly enhance the quality of this article.

Conclusion - Although you conclude and close your review in a good way, there is a more concrete section missing with the limitations of your review, for example, there is a wide difference in some educational contexts and perhaps that influences the teaching-learning relationship, in the same way the comparison between teaching and simulation is a limitation in itself, the differences in the technological methods for eye analysis can also be one, you are the experts on the subject, I recommend you give it another go, it is necessary to talk about the limitations you had so that the next research group that wants to talk about the subject addresses these limitations.

Finally, you carried out very good research and you do not mention concrete practical implications, which could serve to seek an improvement in the learning process, especially for teachers who are more novice. I recommend adding and reviewing an idea.

R.: Thank you for your proposal. We have incorporated it on page 32 (lines 402-431). Additionally, we have included a dedicated subsection outlining the limitations of the included studies on page 29 (lines 377-388).

For reference, please see the following text on page 33 (lines 442-451): “As a systematic review, the Institutional Review Board Statement and Informed Consent Statement do not apply. However, the subsequent experimental scientific studies, which are already underway, have been approved by the Ethics Committee of the University of Coimbra, Faculty of Sports Sciences and Physical Education (reference CE/FCDEF-UC/00142024).

Reviewer 3 Report

Comments and Suggestions for Authors

The paper addresses an important and emerging topic in educational research: the use of eye-tracking technology to analyze teachers' visual behavior. Given the increasing role of technology in education, this research is highly relevant. The study follows a rigorous systematic review methodology (PRISMA criteria) and is registered in the INPLASY database, ensuring transparency and reliability. Covering 41 studies in six major databases, covering a broad range of perspectives and methodologies offers a promissing landscape. Since teachers’ gaze patterns have a potential to impact classroom interactions and instructional effectiveness, the paper has a potential to contribute to teacher training programs.

However, the review acknowledges significant methodological heterogeneity in the selected studies, which may make it difficult to draw generalizable conclusions. To improve this, I suggest widening the introductory or theoretical part for studies which investigated visual attention (used eye-tracking) in particular pedagogy-oriented disciplines. Especially in science education, the outcomes are quite rich.

The paper omits a deeper analysis of potential biases in the included studies, such as publication bias or differences in sample populations. Also, while the study synthesizes findings, the discussion is rather limited. Some sentences are wordy and repetitive, making it harder to follow the discussion. Refining the language for clarity and conciseness would improve readability. From the formal point of view, I noticed wrong apostrophes ' instead of the 9-shaped apostrophes.

The paper mentions the need for "more standardized and robust methodologies" but does not provide concrete recommendations on how to achieve this. In this aspect, the implication should be more robust.

Recommendations:

  1. Set the study in the current context within the use of exe-tracking
  2. Consider grouping the studies into methodological categories to assess which approaches yield the most reliable results.
  3. Discuss potential biases in more detail
  4. Provide an explicit section on limitations of the included studies
  5. Reduce redundancy and streamline sentences for better flow.
  6. Focus on discussion and implications of the study

Author Response

The paper addresses an important and emerging topic in educational research: the use of eye-tracking technology to analyze teachers' visual behavior. Given the increasing role of technology in education, this research is highly relevant. The study follows a rigorous systematic review methodology (PRISMA criteria) and is registered in the INPLASY database, ensuring transparency and reliability. Covering 41 studies in six major databases, covering a broad range of perspectives and methodologies offers a promissing landscape. Since teachers’ gaze patterns have a potential to impact classroom interactions and instructional effectiveness, the paper has a potential to contribute to teacher training programs.

R.: We deeply appreciate the reviewer's comments, which have received our utmost attention. We sincerely thank you for the relevance of your contribution, which will undoubtedly enhance the quality of this article.

Set the study in the current context within the use of exe-tracking AND Consider grouping the studies into methodological categories to assess which approaches yield the most reliable results - However, the review acknowledges significant methodological heterogeneity in the selected studies, which may make it difficult to draw generalizable conclusions. To improve this, I suggest widening the introductory or theoretical part for studies which investigated visual attention (used eye-tracking) in particular pedagogy-oriented disciplines. Especially in science education, the outcomes are quite rich.

R.: Thank you for your proposal. We have incorporated it on page 2 (lines 63-73). Additionally, on page 28 (lines 196-208), while the results are not organized by methodology type, we highlight their heterogeneity.

Discuss potential biases in more detail - The paper omits a deeper analysis of potential biases in the included studies, such as publication bias or differences in sample populations.

R.: Thank you for your proposal. We have incorporated it on page 29 (lines 252-259).

Provide an explicit section on limitations of the included studies - Also, while the study synthesizes findings, the discussion is rather limited.

R.: Thank you for your proposal. We have incorporated it on page 29 (lines 377-388).

Reduce redundancy and streamline sentences for better flow. - Some sentences are wordy and repetitive, making it harder to follow the discussion. Refining the language for clarity and conciseness would improve readability. From the formal point of view, I noticed wrong apostrophes ' instead of the 9-shaped apostrophes.

R.: Thank you for your proposal. We have incorporated it throughout the paper. Additionally, to reduce redundancy and improve readability, we have structured the Discussion section into thematic subsections based on the study's main findings, enabling quicker access to key information.

Focus on discussion and implications of the study - The paper mentions the need for "more standardized and robust methodologies" but does not provide concrete recommendations on how to achieve this. In this aspect, the implication should be more robust.

R.: Thank you for your proposal. We have incorporated it on page 32 (lines 402-431).

For reference, please see the following text on page 33 (lines 442-451): “As a systematic review, the Institutional Review Board Statement and Informed Consent Statement do not apply. However, the subsequent experimental scientific studies, which are already underway, have been approved by the Ethics Committee of the University of Coimbra, Faculty of Sports Sciences and Physical Education (reference CE/FCDEF-UC/00142024).

Round 2

Reviewer 2 Report

Comments and Suggestions for Authors I have nothing further to add to their work; they humbly accepted all suggestions and even improved other aspects. It's a work that contributes to a little-explored line of research and can provide a springboard for future researchers. I'm grateful for the opportunity and congratulations on your work.